# HIGHER ORDER TRANSFORMERS: EFFICIENT ATTENTION MECHANISM FOR TENSOR STRUCTURED DATA

## ABSTRACT

Transformers are now ubiquitous for sequence modeling tasks, but their extension to multi-dimensional data remains a challenge due the to quadratic cost of the attention mechanism. In this paper, we propose Higher-Order Transformers (HOT), a novel architecture designed to efficiently process data with more than two axes, i.e. higher order tensors. To address the computational challenges associated with high-order tensor attention, we introduce a novel Kronecker factorized attention mechanism that reduces the attention cost to quadratic in each axis' dimension, rather than quadratic in the total size of the input tensor. To further enhance efficiency, HOT leverages kernelized attention, reducing the complexity to linear. This strategy maintains the model's expressiveness while enabling scalable attention computation. We validate the effectiveness of HOT on two high-dimensional tasks, including long-term time series forecasting, and 3D medical image classification. Experimental results demonstrate that HOT achieves competitive performance while significantly improving computational efficiency, showcasing its potential for tackling a wide range of complex, multi-dimensional data.

## 1 INTRODUCTION

The Transformer architecture (Vaswani et al., 2017) has revolutionized sequence modeling across various domains, including computer vision (Dosovitskiy et al., 2020), speech recognition (Dong et al., 2018), and reinforcement learning (Parisotto et al., 2020), due to its self-attention mechanism, which effectively captures long-range dependencies and complex patterns in sequential data. However, extending Transformers to handle higher-order data—such as multidimensional arrays or tensors—poses significant challenges due to the quadratic computational and memory costs of the attention mechanism, limiting their application in tasks involving high-dimensional inputs, such as video processing, multidimensional time series forecasting, and 3D medical imaging. High-order data are prevalent in many real-world applications, including climate modeling, which relies on multidimensional time series data capturing temporal and spatial variations (Nguyen et al., 2023); 3D medical imaging, which adds depth to traditional 2D images (Yang et al., 2023); and recommendation systems, where user-item interactions over time and context are modeled as multidimensional tensors (Frolov & Oseledets, 2016). Efficiently processing such data requires models capable of capturing intricate dependencies across multiple dimensions while avoiding prohibitive computational costs.

Several efforts have been made to adapt Transformers for multidimensional data. A common approach is to reshape or flatten the multidimensional input into a sequence (Dosovitskiy et al., 2020), effectively reducing the problem to a one-dimensional case. While this method allows the use of standard Transformers, it disregards the inherent structural information and local dependencies present in the data, as the positional encoding may also fail to communicate this information. Consequently, models may fail to capture essential patterns and exhibit suboptimal performance. Another line of research focuses on applying attention mechanisms along each dimension independently or in a sequential manner. For example, axial attention (Ho et al., 2019) processes data along one axis at a time, reducing computational complexity. As another example, (Song et al., 2016) applies spatial and temporal attention sequentialy. However, this approaches may not fully capture interactions between different dimensions simultaneously, potentially overlooking important cross-dimensional dependencies.

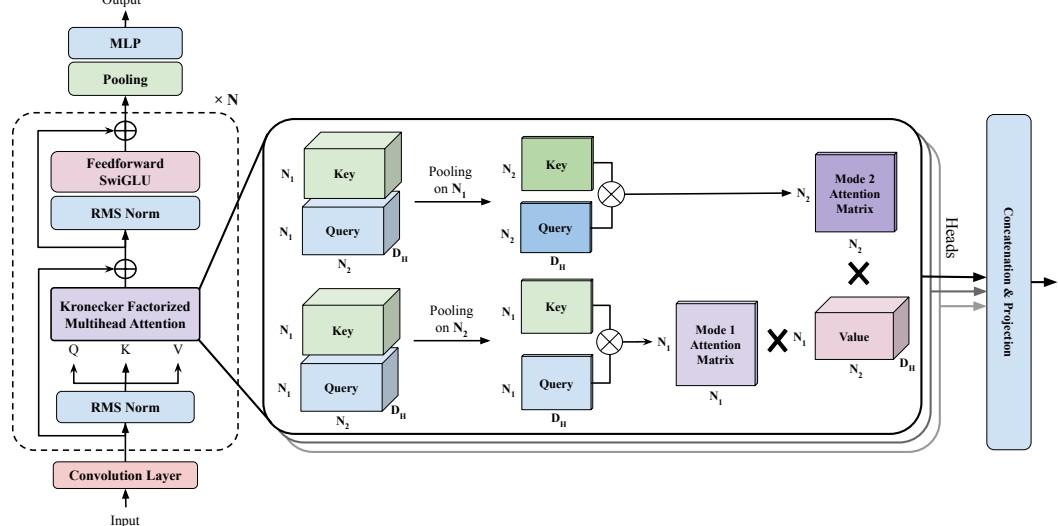

Figure 1: Overall structure of High Order Transformer (HOT) depicting the proposed method for 2D data with size $N_1 \times N_2 \times D$. The model shares the same arrangement with the Transformer encoder while employing Kronecker Factorized Multihead Attention, to reduce the computational complexity. Each mode of the tensor (e.g., $N_1$, $N_2$) has its own attention matrix, and these are combined using the Kronecker product.

In this paper, we introduce **Higher-Order Transformers (HOT)**, a novel architecture designed to efficiently process high-dimensional data represented as tensors. An overall view of our architecture is presented in Figure 1. Our key contributions are as follows:

- We propose a Kronecker decomposition of the high-order attention matrix, significantly reducing the computational complexity.
- To push the boundaries of efficiency, we integrate kernelized attention mechanism into our model reducing the complexity from quadratic to linear with respect to the input size.
- We validate the effectiveness of HOT on two challenging high-dimensional tasks: long-term time series forecasting and 3D medical image classification. In addition, we provide comprehensive ablation study on various aspects of HOT.

**Reproducibility** The code of our method will be made publicly available.

## 2 RELATED WORK

In recent years, various strategies have been developed to make Transformers more efficient for high-dimensional data. One common approach is to flatten the input tensor into a sequence, as in the Vision Transformer (ViT) (Dosovitskiy et al., 2020), which treats image patches as tokens. However, this approach disregards the structural dependencies within the data (Tolstikhin et al., 2021; Lee et al., 2018). To better handle multidimensional structures, axial attention mechanisms like the Axial Transformer (Ho et al., 2019; Wang et al., 2020) apply self-attention along each axis sequentially, reducing complexity but often missing cross-dimensional dependencies crucial for tasks like 3D medical imaging (Hatamizadeh et al., 2022) and climate modeling (Rühling et al., 2022). Similarly, the Sparse Transformer (Child et al., 2019) reduces computation by attending to subsets of the input but struggles with global interactions. Kronecker Attention Networks (Gao et al., 2020) assumes the data to follow matrix-variate normal distributions and accordingly proposes Kronecker attention operators that apply attention on 2D data without flattening. Although the name suggests similarities to our method, it is different from our method as it does not use Kronecker product or decomposition. Tensorized Transformers (Ma et al., 2019) utilize tensor decompositions to reduce memory usage but focus primarily on compression rather than improving cross-dimensional attention. Linear Transformers (Katharopoulos et al., 2020) and Performer (Choromanski et al., 2020) bypass the quadratic softmax bottleneck with linearized attention, making them scalable for long sequences but limited in capturing complex multidimensional relationships. Sparse methods like Longformer (Beltagy et al., 2020) and Reformer (Kitaev et al., 2020) also reduce complexity by restricting attention

to local neighborhoods, but they fail to handle global dependencies in higher-dimensional contexts. Recent works have further improved efficiency and cross-dimensional attention. iTransformer (Liu et al., 2024) optimizes for multivariate time series forecasting by reversing attention across variables, while Crossformer (Zhang & Yan, 2023) uses cross-dimensional attention to capture dependencies between spatial and temporal dimensions, specifically in time series tasks. CdTransformer (Zhu et al., 2024) tackles the challenge of cross-dimensional correlations in 3D medical images with novel attention modules, reducing computational costs and capturing 3D-specific dependencies.

# 3 PRELIMINARIES: TENSOR OPERATIONS

In this section, we introduce key tensor operations that are fundamental to the high-order attention mechanism. Vectors are denoted by lowercase letters (e.g., $v$), matrices by uppercase letters (e.g., $M$), and tensors by calligraphic letters (e.g., $\mathcal{T}$). We use $\otimes$ to represent the Kronecker product and $\times_i$ to denote the tensor product along mode $i$ (Kolda & Bader, 2009). The notation $[k]$ refers to the set $\{1, 2, \ldots, k\}$ for any integer $k$.

**Definition 1** (Tensor). *A $k$-th order tensor $\mathcal{T} \in \mathbb{R}^{N_1 \times N_2 \times \cdots \times N_k}$ generalizes the concept of a matrix to higher dimensions. A tensor can be viewed as a multidimensional array, where each element is indexed by $k$ distinct indices, representing data that varies across $k$ dimensions.*

**Definition 2** (Tensor Mode and Fibers). *A mode-$i$ fiber of a tensor $\mathcal{T}$ is the vector obtained by fixing all indices of $\mathcal{T}$ except the $i$-th one, e.g., $\mathcal{T}_{n_1, n_2, \ldots, n_{i-1}, :, n_{i+1}, \ldots, n_k} \in \mathbb{R}^{N_i}$.*

**Definition 3** (Tensor Slice). *A tensor slice is a two-dimensional section of a tensor, obtained by fixing all but two indices, e.g., $\mathcal{T}_{n_1, n_2, \ldots, n_{i-1}, :, n_{i+1}, \ldots, n_{j-1}, :, n_{j+1}, \ldots, n_k} \in \mathbb{R}^{N_i \times N_j}$*

Slices and fibers extend the familiar concept of matrix rows and columns to higher-dimensional tensors, providing powerful ways to analyze and manipulate multi-way data.

**Definition 4** (Tensor Matricization). *The $i$-th mode matricization of a tensor rearranges the mode-$i$ fibers of the tensor into a matrix. It is denoted as $\mathcal{T}_{(i)} \in \mathbb{R}^{N_i \times (N_1 \cdots N_{i-1} N_{i+1} \cdots N_k)}$.*

**Definition 5** (Mode $n$ tensor product). *The mode $n$ product between a tensor $\mathcal{T} \in \mathbb{R}^{N_1 \times N_2 \times \cdots \times N_k}$ and a matrix $A \in \mathbb{R}^{d \times N_n}$ is denoted by $\mathcal{T} \times_n A \in \mathbb{R}^{N_1 \times N_2 \times \cdots \times N_{n-1} \times d \times N_{n+1} \times \cdots \times N_k}$ and defined by $(\mathcal{T} \times_n A)_{i_1, \cdots, i_k} = \sum_j \mathcal{T}_{i_1, \cdots, i_{n-1}, j, i_{n+1}, \cdots, i_k} A_{i_n, j}$ for all $i_1 \in [N_1], \cdots, i_k \in [N_k]$.*

We conclude this section by stating a useful identity relating matricization, mode $n$ product and the Kronecker product.

**Proposition 1.** *For any tensor $\mathcal{T} \in \mathbb{R}^{N_1 \times N_2 \times \cdots \times N_k \times d}$ of order $k + 1$ and any matrices $A_1 \in \mathbb{R}^{M_1 \times N_1}, \cdots, A_k \in \mathbb{R}^{M_k \times N_k}$, we have $(\mathcal{T} \times_1 A_1 \times_2 A_2 \times_3 \cdots \times_k A_k)_{(k+1)} = \mathcal{T}_{(k+1)}(A_1 \otimes A_2 \otimes \cdots \otimes A_k)^\top$.*

These definitions establish the foundational operations on tensors, which we will build upon to develop the high-order attention mechanism in the next section.

# 4 HIGH ORDER TRANSFORMER

## 4.1 HIGH ORDER ATTENTION

In this section, we first review the self-attention mechanism in Transformer layers (Vaswani et al., 2017), which we extend to higher orders by tensorizing queries, keys, and values, thereby formulating higher order transformer (HOT) layers.

**Standard Scaled Dot-Product Attention** Given an input matrix $X \in \mathbb{R}^{N \times D}$ as an array of $N$ $D$-dimensional embedding vectors, we form the query, key, and value matrices for each attention head $h$ as:

$$Q^h = XW_Q^h, \quad K^h = XW_K^h, \quad V^h = XW_V^h \tag{1}$$

with weight matrices $W_V^h, W_K^h, W_Q^h \in \mathbb{R}^{D \times D_H}$ and output matrix $W_h^O \in \mathbb{R}^{D_H \times D}$, where $D_H$ is the heads' hidden dimension. The standard scaled dot-product attention $g_{\text{Attn}} : \mathbb{R}^{N \times D} \to \mathbb{R}^{N \times D}$ is

Figure 2: Visualization of a rank $R$ Kronecker Decomposition of a high-order full attention matrix $S \in \mathbb{R}^{N_1 N_2 \ldots N_k \times N_1 N_2 \ldots N_k}$ with factor matrices $S_i \in \mathbb{R}^{N_i \times N_i}$. Note that the actual full attention matrix on the LHS can be potentially much larger than what is depicted in the figure.

defined by (Vaswani et al., 2017):

$$g_{\text{Attn}}(X) = \sum_h S^h V^h W_O^h \quad \text{where} \quad S^h = \text{Softmax}\left(\frac{Q^h(K^h)^\top}{\sqrt{D_H}}\right) \tag{2}$$

is the $N \times N$ attention matrix and the Softmax function is applied row-wise.

Although scaled dot-product attention is widely used and has shown great promise across various domains, it comes with limitations that highly impact its scalability. In addition to its quadratic computational complexity, it is originally designed for 1D sequences and can not directly handle higher-order data (e.g., images, videos, etc.) without modification (such as flattening all the dimensions into one). These limitations motivate the development of high-order attention mechanisms that can efficiently handle tensor structured data with multiple *positional* dimensions.

**Generalization to Higher Orders** We now show how the full attention mechanism can be applied to higher-order inputs. Given an input tensor $\mathcal{X} \in \mathbb{R}^{N_1 \times N_2 \times \cdots \times N_k \times D}$, where $N_1, N_2, \ldots, N_k$ are the sizes of the positional modes and $D$ is the hidden dimension, we start by generalizing the attention mechanism to operate over all positional modes collectively. We first compute the query ($\mathcal{Q}$), key ($\mathcal{K}$), and value ($\mathcal{V}$) tensors for each head $h$ by linear projections along the hidden dimension:

$$\mathcal{Q}^h = \mathcal{X} \times_{k+1} (W_Q^h)^\top \in \mathbb{R}^{N_1 \times \cdots \times N_k \times D_H},$$
$$\mathcal{K}^h = \mathcal{X} \times_{k+1} (W_K^h)^\top \in \mathbb{R}^{N_1 \times \cdots \times N_k \times D_H},$$
$$\mathcal{V}^h = \mathcal{X} \times_{k+1} (W_V^h)^\top \in \mathbb{R}^{N_1 \times \cdots \times N_k \times D_H}$$

where $\times_{k+1}$ denotes multiplication along the $(k+1)$-th mode (the hidden dimension).

The scaled dot-product attention scores $S^h \in \mathbb{R}^{(N_1 N_2 \ldots N_k) \times (N_1 N_2 \ldots N_k)}$ are then given by

$$S^h = \text{Softmax}\left(\frac{(\mathcal{Q}_{(k+1)}^h)^\top \mathcal{K}_{(k+1)}^h}{\sqrt{D_H}}\right) \tag{3}$$

where $\mathcal{Q}_{(k+1)}^h \in \mathbb{R}^{(N_1 N_2 \ldots N_k) \times D_H}$ and $\mathcal{K}_{(k+1)} \in \mathbb{R}^{(N_1 N_2 \ldots N_k) \times D_H}$ are the materializations of the query and key tensors, and the Softmax function is again applied row-wise. Each positional index is considered as a single entity in the attention calculation. The output of the high-order attention function $h_{\text{Attn}} : \mathbb{R}^{N_1 \times N_2 \times \cdots \times N_k \times D} \to \mathbb{R}^{N_1 \times N_2 \times \cdots \times N_k \times D}$ is computed by applying the attention weights to the value tensor:

$$(h_{\text{Attn}}(\mathcal{X}))_{(k+1)} = \sum_h (W_O^h)^\top \mathcal{V}_{(k+1)}^h S^h. \tag{4}$$

Lastly, the output is reshaped back to the original tensor shape $N_1 \times N_2 \times \cdots \times N_k \times D$.

While the high-order attention mechanism enables models to capture complex dependencies across multiple dimensions simultaneously, it suffers from significant computational and memory challenges. Specifically, the attention weight tensor scales quadratically with the number of positions, leading to the computational complexity of $\mathcal{O}((N_1 N_2 \ldots N_k)^2)$, which is impractical for large tensors. To address this issue, we propose a low-rank approximation of the high-order attention matrix using a Kronecker product decomposition. This approach dramatically reduces computational complexity while retaining the expressive power of the attention mechanism.

## 4.2 LOW-RANK APPROXIMATION VIA KRONECKER DECOMPOSITION

We parameterize the potentially large high-order attention matrix $S^h \in \mathbb{R}^{(N_1 N_2 ... N_k) \times (N_1 N_2 ... N_k)}$ using a first-order Kronecker decomposition (Figure 2) of the form:

$$S^h \approx S_h^{(1)} \otimes S_h^{(2)} \otimes ... \otimes S_h^{(k)} \tag{5}$$

where each $S_h^{(i)} \in \mathbb{R}^{N_i \times N_i}$ is a factor matrix corresponding to the attention weights over the $i$-th mode for head $h$. Note that having a first-order kronecker decomposition does not mean that $S^h$ is of rank one, as $\mathrm{rank}(S^h) = \prod_i \mathrm{rank}(S_i^h)$. Substituting $S^h$ with its approximation in Eq. (4), we obtain

$$(h_{\mathrm{Attn}}(\mathcal{X}))_{(k+1)} = \sum_{h=1}^{R} (W_O^h)^\top \mathcal{V}_{(k+1)}^h (S_h^{(1)} \otimes S_h^{(2)} \otimes ... \otimes S_h^{(k)}) \tag{6}$$

where each head independently considers one of the modalities.

While not all matrices can be factored into a single Kronecker product as in Eq. (5), we show in Theorem 4.2 below that any attention matrix can be decomposed as a sum of such Kronecker products. The summation across all heads appearing in Eq. (6) functions analogously to a rank $R$ Kronecker decomposition, where the Kronecker rank $R$ correspond to the number of heads. The following theorem shows that a rank $R$ Kronecker decomposition is capable of approximating any high-order attention matrix arbitrarily well, as $R$ increases; ensuring that no significant interactions are missed. This theoretical aspect is crucial for ensuring that the attention mechanism can potentially adapt to any dataset or task requirements.

**Theorem** (Universality of Kronecker decomposition). *Given any high-order attention matrix $S \in \mathbb{R}^{(N_1 N_2 ... N_k) \times (N_1 N_2 ... N_k)}$, there exists an $R \in \mathbb{N}$ such that $S$ can be expressed as a rank $R$ Kronecker decomposition, i.e., $S = \sum_{r=1}^{R} S_r^{(1)} \otimes S_r^{(2)} \otimes ... \otimes S_r^{(k)}$. As $R$ approaches $\min_{j=1,\cdots,k} \prod_{i \neq j} N_i^2$, the approximation is guaranteed to become exact, meaning the Kronecker decomposition is capable of universally representing any high-order attention matrix $S$.*

*Proof.* Proof is presented in the Appendix. $\square$

Now we delve into the computation of the factor matrices $S_h^{(i)}$. As mentioned before, each matrix $S_h^{(i)}$ represents first-order attention weights over the mode $i$. Thus, they can be computed independently using the standard scaled dot-product attention mechanism. Since the input to the attention module is a high-order tensor, computing first-order attention matrices require reshaping of the input query, key, and value tensors. We propose to use a permutation-invariant pooling functions $g_{\mathrm{pool}} : \mathbb{R}^{N_1 \times N_2 \times ... \times N_k \times D_H} \to \mathbb{R}^{N_i \times D_H}$ that takes a high-order tensor as input and only preserves the $i$-th mode and the hidden dimension. In this work, we consider summation over all modes except the $i$-th and last one as the pooling function. We then compute the $i$-th mode attention matrix

$$S_h^{(i)} = \mathrm{Softmax}\left( \frac{\tilde{Q}_i^h (\tilde{K}_i^h)^\top}{\sqrt{D_H}} \right) \tag{7}$$

with pooled matrices $\tilde{Q}_i^h = g_{\mathrm{pool}}(\mathcal{Q}^h) \in \mathbb{R}^{N_i \times D_H}$ and $\tilde{K}_i^h = g_{\mathrm{pool}}(\mathcal{K}^h) \in \mathbb{R}^{N_i \times D_H}$ at a computational cost of $\mathcal{O}(N_i^2 D_H + D_H \prod_j N_j)$.

Explicitly constructing the full attention matrix $S^h = S_h^{(1)} \otimes S_h^{(2)} \otimes ... \otimes S_h^{(k)}$ in Eq. (6) from the factor matrices $S_h^{(i)}$ would negate the computational savings of the Kronecker decomposition. Instead, we exploit properties of the Kronecker product and associative law of matrix and tensor multiplication to apply the attention without forming $S^h$. Formally, it is easy to check that

$$S^h (\mathcal{V}_{(k+1)}^h)^\top = \left( S_h^{(1)} \otimes S_h^{(2)} \otimes ... \otimes S_h^{(k)} \right) (\mathcal{V}_{(k+1)}^h)^\top \tag{8}$$

$$= \left( \left( \left( \mathcal{V}^h \times_1 S_h^{(1)} \right) \times_2 S_h^{(2)} \right) \times_3 ... \times_k S_h^{(k)} \right)_{(k+1)}^\top \tag{9}$$

We can thus multiply each of the attention matrices one by one with the value tensor. The operation on each mode $i$ yields a computational complexity of $\mathcal{O}(N_i D_H (\prod_j N_j))$, resulting in an overall

complexity of $\mathcal{O}(D_H(\sum_i N_i)(\prod_j N_j))$. Thus, for an HOT layer with $R$ heads of width $D_H = D/R$, the total complexity is $\mathcal{O}(D(\sum_i N_i)(\prod_j N_j))$.

While using factorized attention dramatically reduces the computational cost compared to naive high-order attention, the quadratic terms appearing in the final complexity reflect the inherent computational demand of the scaled dot-product attention mechanism, which can itself become substantial for large tensors. To mitigate this final challenge, we integrate kernelized linear attention into the proposed high-order attention mechanism.

### 4.3 LINEAR ATTENTION WITH KERNEL TRICK

Following the work by (Choromanski et al., 2020), we approximate the Softmax function in Eq. (7) using a kernel feature map $\phi : \mathbb{R}^D \to \mathbb{R}^M$:

$$S_h^{(i)} \approx (Z_i^h)^{-1} \phi(\tilde{Q}_i^h) \phi(\tilde{K}_i^h)^\top, \quad (Z_i^h)_{jj} = \phi(\tilde{Q}_i^h)_j \sum_{l=1}^{N_i} \phi(\tilde{K}_i^h)_l^\top. \tag{10}$$

where $Z_i^h \in \mathbb{R}^{N_i \times N_i}$ is the diagonal normalization matrix serving as a normalizing factor. Substituting the Softmax function with Eq. (10) instead of the Softmax in the multiplication between the value tensor $\mathcal{V}^h$ and factor matrix $S_h^{(i)}$ on mode $i$ results in:

$$\mathcal{V}^h \times_i S_h^{(i)} = \mathcal{V}^h \times_i \left( (Z_i^h)^{-1} \phi(\tilde{Q}_i^h) \phi(\tilde{K}_i^h)^\top \right) \tag{11}$$

$$= \left( \left( \mathcal{V}^h \times_i \phi(\tilde{K}_i^h)^\top \right) \times_i \phi(\tilde{Q}_i^h) \right) \times_i (Z_i^h)^{-1} \tag{12}$$

The choice of kernel function $\phi$ is flexible, and we utilize the same kernel function as in (Choromanski et al., 2020), which has been validated both theoretically and empirically. In Eq. (12), we simply used the associative law of matrix multiplication again to reduce the computational complexity of applying a first-order attention matrix on one mode from $\mathcal{O}(N_i D_H(\prod_j N_j))$ to $\mathcal{O}(D_H^2(\prod_j N_j))$ giving us a final complexity of the proposed multi-head factorized high-order attention of $\mathcal{O}(D^2(\prod_j N_j))$. We include the pseudo code for the whole HOT method in Algorithm A.

## 5 EXPERIMENTS

We thoroughly evaluate HOT on two high order data tasks, validating the generality of the proposed framework. At each subsection, we introduce the task, benchmark datasets, and baselines used, and discuss the performance results. Implementation details are presented in the appendix. We close the section by reviewing ablation studies that further confirm our theory and design choices.

### 5.1 LONG-RANGE TIME-SERIES FORECASTING

Given historical observations $X = \{\mathbf{x}_1, \ldots, \mathbf{x}_T\} \in \mathbb{R}^{T \times N}$ with $T$ time steps and $N$ variates, we predict the future $S$ time steps $Y = \{\mathbf{x}_{T+1}, \ldots, \mathbf{x}_{T+S}\} \in \mathbb{R}^{S \times N}$.

**Datasets** We extensively include 5 real-world datasets in our experiments, including ECL, Exchange, Traffic, Weather used by Autoformer Wu et al. (2021), and Solar-Energy proposed in LSTNet Lai et al. (2017). Further dataset details are in the Appendix.

**Baselines** We choose 11 well-acknowledged forecasting models as our benchmark, including (1) Transformer-based methods: ITransformer Liu et al. (2024), Crossformer Zhang & Yan (2023), Autoformer Wu et al. (2021), FEDformer Zhou et al. (2022), Stationary Liu et al. (2023), PatchTST Nie et al. (2023); (2) Linear-based methods: DLinear Zeng et al. (2022), TiDE Das et al. (2024), RLinear Li et al. (2023); and (3) TCN-based methods: SCINet Liu et al. (2021), TimesNet Wu et al. (2023).

**Results** Comprehensive forecasting results are provided in Table 1, with the best results highlighted in bold and the second-best underlined. Lower MSE/MAE values reflect more accurate predictions. As seen in the table, our proposed method, HOT, outperforms all baseline models across all datasets, achieving the best MSE and MAE scores in every case. Specifically, HOT provides significant improvements on larger, more complex datasets such as ECL and Traffic, where capturing multivariate

Table 1: Multivariate forecasting results with prediction lengths $S \in \{96, 192, 336, 720\}$ and fixed lookback length $T = 96$ with the best in **Bold** and second-best in underline. Results are averaged from all prediction lengths. HOT outperforms all baselines over all the datasets while having a low computational complexity. Full results are available in Table 12 in the appendix.

| Models | Params | Complexity | ECL | | Weather | | Traffic | | Solar | | Exchange | |
|---|---|---|---|---|---|---|---|---|---|---|---|---|
| | | | MSE | MAE | MSE | MAE | MSE | MAE | MSE | MAE | MSE | MAE |
| AutoFormer (2021) | 15M | $\mathcal{O}(NTlogT)$ | 0.227 | 0.338 | 0.338 | 0.382 | 0.628 | 0.379 | 0.885 | 0.711 | 0.613 | 0.539 |
| SCINet (2022) | - | $\mathcal{O}(NTlogT)$ | 0.268 | 0.365 | 0.292 | 0.363 | 0.804 | 0.509 | 0.282 | 0.375 | 0.750 | 0.626 |
| FedFormer (2022) | 21M | $\mathcal{O}(NTlogT)$ | 0.214 | 0.327 | 0.309 | 0.360 | 0.610 | 0.376 | 0.291 | 0.381 | 0.519 | 0.429 |
| Stationary (2022) | - | $\mathcal{O}(NTD)$ | 0.193 | 0.296 | 0.288 | 0.314 | 0.624 | 0.340 | 0.261 | 0.381 | 0.461 | 0.454 |
| TiDE (2023) | - | $\mathcal{O}(NTD)$ | 0.251 | 0.344 | 0.271 | 0.320 | 0.760 | 0.473 | 0.347 | 0.417 | 0.370 | 0.417 |
| Crossformer (2023) | - | $\mathcal{O}(NTlogT + N^2T)$ | 0.244 | 0.334 | 0.259 | 0.315 | 0.550 | 0.304 | 0.641 | 0.639 | 0.940 | 0.707 |
| RLinear (2023) | - | $\mathcal{O}(NT)$ | 0.219 | 0.298 | 0.272 | 0.291 | 0.626 | 0.378 | 0.369 | 0.356 | 0.378 | 0.417 |
| DLinear (2023) | 140K | $\mathcal{O}(NT)$ | 0.212 | 0.300 | 0.265 | 0.317 | 0.625 | 0.383 | 0.330 | 0.401 | 0.354 | 0.414 |
| TimesNet (2023) | 301M | $\mathcal{O}(NTlogT)$ | 0.192 | 0.295 | 0.259 | 0.287 | 0.620 | 0.336 | 0.301 | 0.319 | 0.416 | 0.443 |
| PatchTST (2023) | 1.5M | $\mathcal{O}(NTD)$ | 0.205 | 0.290 | 0.259 | 0.281 | 0.481 | 0.304 | 0.270 | 0.307 | 0.367 | 0.404 |
| iTransformer (2024) | 3M | $\mathcal{O}(N^2D + NTD)$ | 0.178 | 0.270 | 0.258 | 0.278 | 0.428 | 0.282 | 0.233 | 0.262 | 0.360 | 0.403 |
| HOT (Ours) | 620K | $\mathcal{O}(NTD^2)$ | **0.172** | **0.268** | **0.250** | **0.275** | **0.422** | **0.280** | **0.229** | **0.256** | **0.343** | **0.394** |

dependencies is critical. For smaller datasets like Exchange and Weather, HOT also outperforms baselines, but the gap between HOT and other models, like iTransformer, is narrower, which could be due to the smaller number of variates in these datasets, reducing the advantage of higher-order attention. Moreover, while other high-performing models like iTransformer and PatchTST deliver competitive results, they come with higher computational complexities. For example, iTransformer, with a complexity of $\mathcal{O}(N^2D + NTD)$, scales poorly with the number of time steps and variates, making it less efficient for large datasets. In contrast, HOT maintains a lower complexity of $\mathcal{O}(NTD^2)$, which scales better with both dimensions and time, especially for higher-dimensional data. This efficiency is particularly important for large datasets such as ECL and Traffic, where HOT balances performance and computational cost, outperforming even models like FedFormer and Crossformer, which have similar or higher complexity. Overall, HOT not only achieves superior accuracy but also offers improved scalability and efficiency for multivariate time series forecasting tasks with much fewer parameters.

## 5.2 3D Medical Image Classification

Given a 3D image $\mathcal{X} \in \mathbb{R}^{W \times H \times D}$ with width $W$, height $H$, and depth $D$, we predict the image class probability $y \in \mathbb{R}^C$ over a set of $C$ classes.

**Dataset** MedMNIST v2 Yang et al. (2023) is a large-scale benchmark for medical image classification on standardized MNIST-like 2D and 3D images with diverse modalities, dataset scales, and tasks. We primarily experiment on the 3D portion of MedMNIST v2, namely the Organ, Nodule, Fracture, Adrenal, Vessel, and Synapse datasets. The size of each image is $28 \times 28 \times 28$ (3D).

**Baselines** We choose 11 medical image classifier models including ResNet-18/ResNet-50 (He et al., 2015) with 2.5D / 3D / ACS (Yang et al., 2021) convolutions (Yang et al., 2023), DWT-CV (Cheng et al., 2022), Auto-Keras, and Auto-sklearn (Yang et al., 2023), MDANet (Huang et al., 2022), and CdTransformer (Zhu et al., 2024).

**Results** The results presented in Table 2 demonstrate the superior performance of our Higher Order Transformer (HOT) across multiple medical imaging datasets. HOT achieves the highest accuracy and AUC on Organ, Fracture, Adrenal, and Vessel datasets, and the second-best performance in Synapse and Nodule showcasing its robust classification capabilities. While models like CdTransformer achieve better performance on Nodule and Synapse, they do so with a significantly increased computational complexity of $\mathcal{O}(N^3D^2 + N^4D)$ compared to HOT's $\mathcal{O}(N^3D^2)$. Additionally, HOT consistently outperforms other state-of-the-art methods such as DWT-CV and MDANet over both metrics, balancing high performance with lower computational demands and much fewer parameters.

## 5.3 Ablation Study

To verify the rational business of the proposed HOT, we provide detailed ablations covering analyses on rank of the attention factorization, attention order, attention type, and lastly, individual model components namely high-order attention module and the feed-forward module.

Table 2: 3D image classification results on MedMNIST3D with the best in **Bold** and second-best in underline. Our method (HOT) achieves top performance across several datasets while maintaining a low computational complexity with respect to the input size ($N$) and hidden dimension size ($D$).

| Models | Params | Complexity | Organ | | Nodule | | Fracture | | Adrenal | | Vessel | | Synapse | |
|---|---|---|---|---|---|---|---|---|---|---|---|---|---|---|
| | | | AUC | ACC | AUC | ACC | AUC | ACC | AUC | ACC | AUC | ACC | AUC | ACC |
| ResNet-18+2.5D (2016) | 12M | $\mathcal{O}(N^3 D^2)$ | 97.7 | 78.8 | 83.8 | 83.5 | 58.7 | 45.1 | 71.8 | 77.2 | 74.8 | 84.6 | 63.4 | 69.6 |
| ResNet-18+3D (2016) | 12M | $\mathcal{O}(N^3 D^2)$ | 99.6 | 90.7 | 86.3 | 84.4 | 71.2 | 50.8 | 82.7 | 72.1 | 87.4 | 87.7 | 82.0 | 74.5 |
| ResNet-18+ACS (2016) | 12M | $\mathcal{O}(N^3 D^2)$ | 99.4 | 90.0 | 87.3 | 84.7 | 71.4 | 49.7 | 83.9 | 75.4 | 93.0 | 92.8 | 70.5 | 72.2 |
| ResNet-50+2.5D (2016) | 26M | $\mathcal{O}(N^3 D^2)$ | 97.4 | 76.9 | 83.5 | 84.8 | 55.2 | 39.7 | 73.2 | 76.3 | 75.1 | 87.7 | 66.9 | 73.5 |
| ResNet-50+3D (2016) | 26M | $\mathcal{O}(N^3 D^2)$ | 99.4 | 88.3 | 87.5 | 84.7 | 72.5 | 49.4 | 82.8 | 74.5 | 90.7 | 91.8 | 85.1 | 79.5 |
| ResNet-50+ACS (2023) | 26M | $\mathcal{O}(N^3 D^2)$ | 99.4 | 88.9 | 88.6 | 84.1 | **75.0** | 51.7 | 82.8 | 75.8 | 91.2 | 85.8 | 71.9 | 70.9 |
| Auto-sklearn (2015) | - | - | 97.7 | 81.4 | 91.4 | 87.4 | 62.8 | 45.3 | 82.8 | 80.2 | 91.0 | 91.5 | 63.1 | 73.0 |
| Auto-Keras (2019) | - | - | 97.9 | 80.4 | 84.4 | 83.4 | 64.2 | 45.8 | 80.4 | 70.5 | 77.3 | 89.4 | 53.8 | 72.4 |
| DWT-CV (2022) | 12-25M | $\mathcal{O}(N^3 D^2)$ | 99.4 | 91.2 | 91.2 | **91.2** | 72.3 | 53.1 | 86.6 | 81.2 | 90.5 | 91.2 | - | - |
| MDANet (2022) | 7M | $\mathcal{O}(N^3 D^2)$ | 98.9 | 89.7 | 86.8 | 86.00 | - | - | 83.9 | 81.5 | 90.1 | 92.9 | 71.2 | 75.0 |
| CdTransformer (2024) | - | $\mathcal{O}(N^3 D^2 + N^4 D)$ | - | - | **94.3** | 90.3 | 72.4 | 52.9 | 88.4 | 83.6 | 95.9 | 92.9 | **87.9** | **83.2** |
| HOT (Ours) | 7M | $\mathcal{O}(N^3 D^2)$ | **99.7** | **94.0** | 90.8 | 87.7 | 73.2 | **57.5** | **88.6** | **85.2** | **97.6** | **96.3** | 83.4 | 80.9 |

### 5.3.1 ATTENTION RANK

In this section, we evaluate the impact of the number of attention heads on the performance of HOT across both medical imaging and time series datasets. The attention rank, governed by the number of heads, plays a critical role in the model's ability to capture diverse patterns across multiple dimensions by approximating the original high-order attention. We conduct ablation experiments by varying the number of heads to observe how it affects model accuracy and error rates. For 3D medical image datasets (Figure 3 Left), increasing the number of attention heads initially improves accuracy, however, after a certain threshold, performance declines. This drop is due to the fixed hidden dimension, which causes the dimension of each head to decrease as the number of heads increases, reducing each head's ability to capture rich features and leading to less expressive attention mechanisms. For the time series datasets (Figure 3 Right), the use of more heads improves performance, with the MSE consistently decreasing as the number of heads increases.

Although the effective rank for the Kronecker decomposition varies for different tasks and datasets and is highly dependent on the data size and its characteristics, in practice as shown in Figures 3, we see that low-rank approximation (where the rank or the number of heads is not a large number) is expressive enough to yield good results. To that end, we treat the rank as a hyper-parameter of the model and choose it by hyper-parameter search based on validation metrics for each dataset. This approach makes it quite easy to achieve a proper performance, without any need for exhaustive rank analysis on the data.

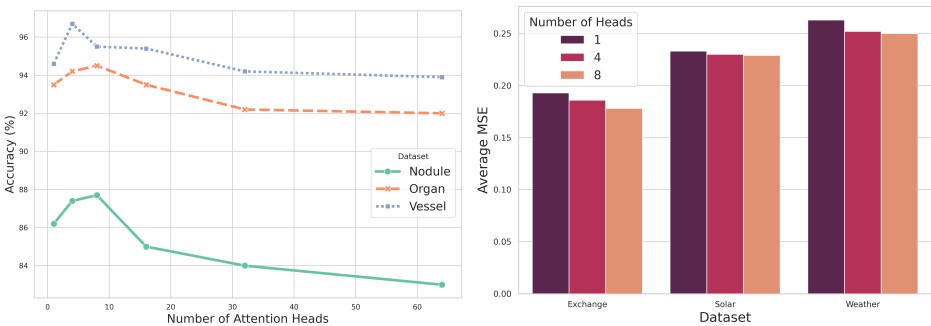

Figure 3: Effect of increasing attention heads on model performance. **Left:** 3D medical image datasets, **Right:** For multivariate timeseries datasets.

### 5.3.2 ATTENTION ORDER

We conducted an ablation study to explore the effects of increasing the attention order on the performance of our proposed HOT. The attention order refers to the number of dimensions over which attention is applied, extending beyond traditional sequence-based attention to handle high-dimensional data effectively. We evaluate performance on both time series forecasting and 3D medical image classification tasks under different attention configurations. As shown in Tables 3

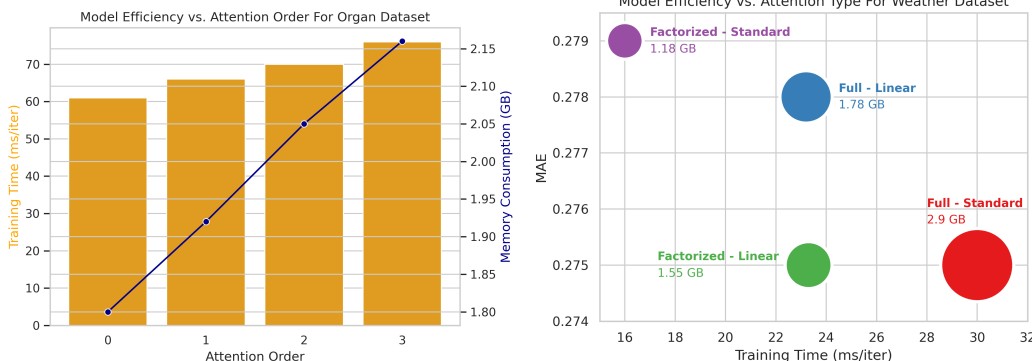

Figure 4: **Left:** Memory footprint and training time of HOT under various attention orders. **Right:** Performance vs. training time of HOT under various attention types. The size of each circle indicates its relative memory consumption. Note that the number of model parameters is fixed for each dataset.

and 4, applying higher-order attention consistently improves the performance across all datasets, outperforming configurations with lower-order attention. However, the training time and memory consumption of the model increases linearly with the order of the attention as depicted in Figure 4 (Left). It is worth mentioning that the model with no attention is equivalent to an MLP with residual connections and normalizations. Importantly, all models maintain the same number of parameters and the same computational and memory complexity for each dataset, highlighting that the performance gains are attributable to the increased attention order without adding. Details of memory consumption and training time are presented in the appendix.

Table 3: Quantitative evaluation of timeseries forecasting performance under different attention orders. Results are averaged from all prediction lengths. Higher-order attention outperforms lower-order attention while maintaining the same computational complexity. HOT is highlighted in green.

| Attention Dimensions | | Complexity | Solar | | Weather | | Electricity | |
|:---:|:---:|:---:|:---:|:---:|:---:|:---:|:---:|:---:|
| Variable | Time | | MSE | MAE | MSE | MAE | MSE | MAE |
| ✗ | ✗ | $\mathcal{O}(NTD^2)$ | 0.240 | 0.269 | 0.265 | 0.283 | 0.193 | 0.288 |
| ✓ | ✗ | $\mathcal{O}(NTD^2)$ | 0.237 | 0.262 | 0.257 | 0.282 | 0.184 | 0.284 |
| ✗ | ✓ | $\mathcal{O}(NTD^2)$ | 0.235 | 0.260 | 0.254 | 0.278 | 0.180 | 0.272 |
| ✓ | ✓ | $\mathcal{O}(NTD^2)$ | **0.229** | **0.256** | **0.250** | **0.275** | **0.172** | **0.268** |

Table 4: Quantitative evaluation of 3D image classification performance under different attention orders. Higher-order attention outperforms lower-order attention with the same computational complexity. HOT is highlighted in green.

| Attention Dimensions | | | Complexity | Organ | | Nodule | | Vessel | |
|:---:|:---:|:---:|:---:|:---:|:---:|:---:|:---:|:---:|:---:|
| Height | Width | Depth | | AUC | ACC | AUC | ACC | AUC | ACC |
| ✗ | ✗ | ✗ | $\mathcal{O}(N^3D^2)$ | 99.4 | 90.1 | 81.9 | 83.7 | 88.7 | 94.2 |
| ✓ | ✗ | ✗ | $\mathcal{O}(N^3D^2)$ | 99.6 | 92.2 | 82.1 | 84.8 | 90.9 | 95.6 |
| ✓ | ✓ | ✗ | $\mathcal{O}(N^3D^2)$ | 99.6 | 93.9 | 84.3 | 85.8 | 91.4 | 96.0 |
| ✓ | ✓ | ✓ | $\mathcal{O}(N^3D^2)$ | **99.7** | **94.0** | **90.8** | **87.7** | **97.6** | **96.3** |

### 5.3.3 ATTENTION TYPE

In this section, we compare naive full attention with the proposed factorized attention approach. To implement the full high-order attention, we flatten all dimensions except for the batch and hidden dimensions, then apply standard scaled dot-product attention to the flattened input. Finally, the output is reshaped back to its original dimensions. As discussed earlier, the quadratic complexity of standard attention results in significantly higher computational and memory requirements. To address this,

we also incorporated kernelized attention into the naive full attention, reducing its complexity to linear—matching the efficiency of our proposed method. As shown in Tables 6 and 5, with proper hyperparameter tuning, our proposed HOT (highlighted in green) achieves results comparable to standard full attention while requiring far less computation, memory, and training time.

An interesting observation from the results is that combining factorization with the kernel trick improves performance compared to factorization alone. This is achieved by introducing additional inductive bias into the attention computation. While linear attention without factorization performs similarly to the combined approach in 3D medical image classification tasks, factorization offers several distinct advantages: 1. It enables the control of model expressivity by adjusting the rank based on the computational cost. 2. It enables the explanation of large high-order attentions, which are otherwise infeasible to analyze, using smaller first-order attention matrices that are already well-studied and interpretable. 3. It allows flexible treatment of attention across different dimensions by enabling the application of independent attention masks on each axis without interference. 4. When combined with linear attention, it results in lower MAE and reduced memory consumption compared to linear attention alone as shown in Figure 4 (Right).

Table 5: Quantitative Analysis for the effect of applying Kronecker factorization and kernel trick in attention on HOT performance for timeseries forecasting. Results are averaged from all prediction lengths. HOT is highlighted in green.

| Attention Setting | | Complexity | Solar | | Weather | | Electricity | |
|---|---|---|---|---|---|---|---|---|
| Factorized | Linear | | MSE | MAE | MSE | MAE | MSE | MAE |
| ✗ | ✗ | $\mathcal{O}(N^2T^2D)$ | **0.219** | 0.257 | **0.245** | **0.275** | **0.162** | 0.261 |
| ✗ | ✓ | $\mathcal{O}(NTD^2)$ | 0.229 | 0.261 | 0.248 | 0.278 | 0.171 | 0.270 |
| ✓ | ✗ | $\mathcal{O}(NTD(N+T))$ | 0.236 | 0.267 | 0.258 | 0.279 | 0.187 | 0.280 |
| ✓ | ✓ | $\mathcal{O}(NTD^2)$ | 0.229 | **0.256** | 0.250 | **0.275** | 0.172 | 0.268 |

Table 6: Quantitative Analysis for the effect of applying Kronecker factorization and kernel trick in attention on HOT performance for 3D image classification. HOT is highlighted in green.

| Attention Setting | | Complexity | Organ | | Nodule | | Vessel | |
|---|---|---|---|---|---|---|---|---|
| Factorized | Linear | | AUC | ACC | AUC | ACC | AUC | ACC |
| ✗ | ✗ | $\mathcal{O}(N^6D)$ | 99.6 | 92.6 | **91.2** | **87.8** | 97.7 | **96.6** |
| ✗ | ✓ | $\mathcal{O}(N^3D^2)$ | **99.7** | **94.0** | 89.7 | 87.7 | **98.1** | 96.3 |
| ✓ | ✗ | $\mathcal{O}(N^4D)$ | 99.5 | 91.3 | 86.1 | 86.4 | 97.2 | 96.1 |
| ✓ | ✓ | $\mathcal{O}(N^3D^2)$ | **99.7** | **94.0** | 90.8 | 87.7 | 97.6 | 96.3 |

## 6 CONCLUSION

In this paper, we addressed the challenge of extending Transformers to high-dimensional data, often limited by the quadratic cost of attention mechanisms. While methods like flattening inputs or sparse attention reduce computational overhead, they miss essential cross-dimensional dependencies and structural information. We introduced Higher-Order Transformers (HOT) with Kronecker factorized attention to lower complexity while preserving expressiveness. This approach processes high-dimensional data efficiently, with complexity scaling quadratically per dimension. We further integrated kernelized attention for additional scalability and a complexity scaling linearly per dimension. HOT demonstrated strong performance in tasks such as time series forecasting and 3D medical image classification, proving both its effectiveness and efficiency. Future work could enhance HOT's interpretability by analyzing attention maps or exploring alternative pooling methods for better information aggregation. Additionally, adapting HOT as an autoregressive model could enhance spatial and temporal coherency for generative tasks like video synthesis and climate forecasting. Lastly, although the use of the proposed factorized linear attention was only studied in the context of the Transformers architecture in this work, its capabilities in other architectures such as Attention U-Net Oktay et al. (2018) is yet to be explored.

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

## A  HOT ALGORITHM

---

**Algorithm 1** High-Order Attention with Kronecker Decomposition and Linear Approximation

---

**Require:** Input tensor $\mathcal{X} \in \mathbb{R}^{N_1 \times N_2 \times \cdots \times N_k \times D}$, Number of heads $H$, Hidden dimension $D_H$, Kernel feature map $\phi$

**Ensure:** Output tensor $\mathcal{Y} \in \mathbb{R}^{N_1 \times N_2 \times \cdots \times N_k \times D}$

1: Initialize query, key, value projection matrices $W_Q^h, W_K^h, W_V^h \in \mathbb{R}^{D \times D_H}$, output projection matrices $W_O^h \in \mathbb{R}^{D_H \times D}$

2: Initialize empty output tensor $\mathcal{Y} \in \mathbb{R}^{N_1 \times N_2 \times \cdots \times N_k \times D} \leftarrow 0$

3: **for** each head $h = 1$ to $H$ **do**

4:     Compute query, key, and value tensors:

$$\mathcal{Q}^h = \mathcal{X} \times_{k+1} W_Q^h, \quad \mathcal{K}^h = \mathcal{X} \times_{k+1} W_K^h, \quad \mathcal{V}^h = \mathcal{X} \times_{k+1} W_V^h$$

5:     Initialize output tensor $\mathcal{P} \leftarrow \mathcal{V}^h$

6:     **for** each mode $i = 1$ to $k$ **do**

7:         Pool the query and key tensors across all modes except $i$:

$$\tilde{Q}_i^h = g_{\text{pool}}(\mathcal{Q}^h), \quad \tilde{K}_i^h = g_{\text{pool}}(\mathcal{K}^h)$$

8:         Compute first-order attention matrix for mode $i$ with kernel trick:

$$S_h^{(i)} \approx (Z_i^h)^{-1} \phi(\tilde{Q}_i^h) \phi(\tilde{K}_i^h)^\top$$

9:         Apply the Kronecker attention to $\mathcal{P}$:

$$\mathcal{P} = \mathcal{P} \times_i S_h^{(i)}$$

10:     **end for**

11:     Update the output tensor:

$$\mathcal{Y} = \mathcal{Y} + \mathcal{P}^h \times_{k+1} W_O^h$$

12: **end for**

13: **return** $\mathcal{Y}$

---

## B  UNIVERSALITY OF THE KRONECKER DECOMPOSITION

**Theorem** (4.2). *Given any high-order attention matrix $S \in \mathbb{R}^{(N_1 N_2 ... N_k) \times (N_1 N_2 ... N_k)}$, there exists an $R \in \mathbb{N}$ such that $S$ can be expressed as a rank $R$ Kronecker decomposition, i.e., $S = \sum_{r=1}^R S_r^{(1)} \otimes S_r^{(2)} \otimes ... \otimes S_r^{(k)}$. As $R$ approaches $\min_{j=1,\cdots,k} \prod_{i \neq j} N_i^2$, the approximation is guaranteed to become exact, meaning the Kronecker decomposition is capable of universally representing any high-order attention matrix $S$.*

*Proof.* Let $\mathcal{S}$ be the tensor obtained by reshaping the attention matrix $S$ into a tensor of size $N_1 \times N_2 \times \cdots \times N_k \times N_1 \times N_2 \times \cdots \times N_k$ and let $\mathcal{T} \in \mathbb{R}^{N_1^2 \times N_2^2 \times \cdots \times N_k^2}$ be the tensor obtained by merging each pair of modes corresponding to one modality[1]. Let $R$ be the CP rank of $\mathcal{T}$ and let $\mathcal{T} = \sum_{r=1}^R s_r^{(1)} \circ s_r^{(2)} \circ \cdots \circ s_r^{(k)}$ be a CP decomposition, where $\circ$ denotes the outer product and each $s_r^{(i)} \in \mathbb{R}^{N_i^2}$ for $i = 1, \cdots, k$ (see, e.g., (Kolda & Bader, 2009) for an introduction to the CP decomposition). By reshaping each $s_r^{(i)} \in \mathbb{R}^{N_i^2}$ into a matrix $S_r^{(i)} \in \mathbb{R}^{N_i \times N_i}$, one can check that

$$\mathcal{S}_{i_1,\cdots,i_k,j_1,\cdots,j_k} = \sum_{r=1}^R (S_r^{(1)})_{i_1,j_1} \otimes (S_r^{(2)})_{i_2,j_2} \otimes \cdots \otimes (S_r^{(k)})_{i_k,j_k}$$

from which it follows that $S = \sum_{r=1}^R S_r^{(1)} \otimes S_r^{(2)} \otimes ... \otimes S_r^{(k)}$, as desired.

---

[1]i.e., in pytorch $\mathcal{T}$ would be obtained obtained by permuting the modes of $\mathcal{S}$ and reshaping: $\text{torch.transpose}(\mathcal{S}, [0, k+1, 1, k+2, \cdots, k-1, 2k-1].\text{reshape}([N_1^2, \cdots, N_k^2])$

The second part of the theorem comes from the fact that $\min_{i=1,\cdots,p} \prod_{j\neq i} d_j$ is a well known upper bound on the CP rank of a tensor of shape $d_1 \times d_2 \times \cdots \times d_k$ (see again (Kolda & Bader, 2009)). $\square$

It is worth noting that based on the given proof, the CP decomposition of the attention matrix reshaped into a tensor of size $N_1^2 \times N_2^2 \times \cdots \times N_k^2$ is *exactly equivalent* to the Kronecker factorization of the attention matrix used in HOT.

## C  DATASETS DETAILS

### C.1  LONG-RANGE TIME-SERIES FORECASTING

We evaluate the performance of the proposed HOT model on seven real-world datasets: (1) Exchange (Wu et al., 2021), which contains daily exchange rates for eight countries from 1990 to 2016, (2) Weather (Wu et al., 2021), consisting of 21 meteorological variables recorded every 10 minutes in 2020 at the Max Planck Biogeochemistry Institute, (3) ECL (Wu et al., 2021), which tracks hourly electricity consumption for 321 clients, (4) Traffic (Wu et al., 2021), collecting hourly road occupancy data from 862 sensors on San Francisco Bay area freeways between January 2015 and December 2016, and (5) Solar-Energy (Lai et al., 2017), recording solar power production from 137 photovoltaic (PV) plants, sampled every 10 minutes in 2006.

We follow the data processing and train-validation-test split protocol used in TimesNet (Wu et al., 2023), ensuring datasets are chronologically split to prevent any data leakage. For forecasting tasks, we use a fixed lookback window of 96 time steps for the Weather, ECL, Solar-Energy, and Traffic datasets, with prediction lengths of 96, 192, 336, 720. Further dataset details are presented in Table 7.

Table 7: Timeseries forecasting dataset details.

| Dataset | Variables | Prediction Length | Train/Val/Test Size | Sample Frequency |
|---------|-----------|-------------------|---------------------|------------------|
| Exchange | 8 | {96, 192, 336, 720} | (5120, 665, 1422) | Daily |
| Weather | 21 | {96, 192, 336, 720} | (36792, 5271, 10540) | 10min |
| ECL | 321 | {96, 192, 336, 720} | (18317, 2633, 5261) | Hourly |
| Traffic | 862 | {96, 192, 336, 720} | (12185, 1757, 3509) | Hourly |
| Solar-Energy | 137 | {96, 192, 336, 720} | (36601, 5161, 10417) | 10min |

### C.2  3D MEDICAL IMAGE CLASSIFICATION

We conduct experiments on the 3D subset of the Medical MNIST dataset (Yang et al., 2023). All datasets have an image size of $28 \times 28 \times 28$ voxels, allowing for consistent 3D image classification across different medical domains. The images come from various sources, ranging from human CT scans to animal microscopy, and have been adapted to create challenging classification tasks. Details are presented in Table 8.

- OrganMNIST3D is based on the same CT scan data used for the Organ{A,C,S}MNIST datasets, but instead of 2D projections, it directly uses the 3D bounding boxes of 11 different body organs. The dataset is adapted for a multiclass classification on organ identification from volumetric medical data.

- NoduleMNIST3D originates from the LIDC-IDRI dataset, a public repository of thoracic CT scans designed for lung nodule segmentation and malignancy classification. For this study, the dataset has been adapted for binary classification of lung nodules based on malignancy levels, excluding cases with indeterminate malignancy. The images are center-cropped and spatially normalized to retain a consistent voxel spacing.

- AdrenalMNIST3D features 3D shape masks of adrenal glands collected from patients at Zhongshan Hospital, Fudan University. Each shape is manually annotated by an expert endocrinologist using CT scans, though the original scans are not included in the dataset to

protect patient privacy. Instead, the dataset focuses on binary classification of normal versus abnormal adrenal glands based on the processed 3D shapes derived from the scans.

- FractureMNIST3D is derived from the RibFrac dataset, which contains CT scans of rib fractures. The dataset classifies rib fractures into three categories (buckle, nondisplaced, and displaced), omitting segmental fractures due to the resolution of the images.

- VesselMNIST3D uses data from the IntrA dataset, which includes 3D models of intracranial aneurysms and healthy brain vessels reconstructed from magnetic resonance angiography (MRA) images. The dataset focuses on classifying healthy vessel segments versus aneurysms, with the models voxelized into 3D volumes.

- SynapseMNIST3D is based on high-resolution 3D electron microscopy images of a rat's brain, with the dataset focusing on classifying synapses as either excitatory or inhibitory. The data were annotated by neuroscience experts, and each synapse is cropped from the original large-scale volume and resized.

Table 8: MedMNIST 3D datasets details.

| Dataset | Modality | Number of Classes | Train/Val/Test Size |
|---------|----------|-------------------|---------------------|
| Organ | Abdominal CT | 11 | (972, 161, 610) |
| Nodule | Chest CT | 2 | (1158, 165, 310) |
| Adrenal | Shape from Abdominal CT | 2 | (1188, 98, 298) |
| Fracture | Chest CT | 3 | (1027, 103, 240) |
| Vessel | Shape from Brain MRI | 2 | (1335, 192, 382) |
| Synapse | Electron Microscope | 2 | (1230, 177, 352) |

## D  IMPLEMENTATION DETAILS

Table 9: Hyperparameter Search Space.

| Hyperparameter | Value List |
|----------------|------------|
| Number of HOT Blocks | [1, 2, 3, 4] |
| Number of Hidden Dimensions | [64, 128, 256] |
| Dropout | [0, 0.1, 0.2, 0.3, 0.4] |
| Number of Attention Heads | [1, 4, 8] |
| Convolution Kernel Size (3D Image Classification) | [3, 5, 7] |
| Convolution Kernel Size (TimeSeries Forecasting) | [1, 4, 8, 16] |
| Pooling Function | [Mean, Flatten] |

**Timeseries Forecasting**  The convolution encoder is a single 1D convolution layer with kernel size and stride both set to 4 applied on the temporal axis. This is equal to dividing the input timeseries into patches of size 4 and applying a linear projection to the hidden space of the model. Rotary positional encoding Su et al. (2021) is used only for the time axis. The output of the transformer is pooled before being fed to the final MLP layer by either taking the average or flattening. We conduct forecasting experiments by training models on each dataset. Following the same split of training/validation/test sets as in Liu et al. (2024), the model weights from the epoch with the lowest MAE on the validation set are selected for comparison on the test set.

**3D Medical Image Classification**  The convolution encoder is implemented as a multilayer 3D convolution with a total downsampling by a factor of 4, while Rotary positional encoding is used for all three spatial dimensions. The output of the transformer is pooled before being fed to the final

MLP classifier by either taking the average or flattening. We conduct classification experiments by training models on each dataset. Following the official split of training/validation/test sets, we train all models on the training sets for 100 epochs. The model weights from the epoch with the highest AUC score on the validation set are selected for comparison on the test set.

We fixed the feature map function in the linear attention to *SMReg* from Performer Choromanski et al. (2020) for all our models, as it was shown to be the most stable, fast converging and expressive by the authors. All the experiments are implemented in PyTorch and conducted on a single NVIDIA A100 GPU. We utilize ADAM (Kingma & Ba, 2017) with an initial learning rate of $2 \times 10^{-4}$ and L2 loss for the timeseries forecasting task and cross-entropy loss for the medical image classification task. The batch size is uniformly set to 32 and the number of training epochs is fixed to 100. We conduct hyperparameter tuning based on the search space shown in Table 9.

# E  ADDITIONAL RESULTS

We report the standard deviation of HOT performance over five runs with different random seeds in Tables 10 and 11, which exhibits that the performance of HOT is stable on both tasks. We also report symmetric MAPE (SMAPE) for timeseries forecasting in Table 11.

Table 10: Robustness of HOT performance on 3D medical image classification. The results are obtained from five random seeds.

| Organ | | Nodule | | Fracture | |
|---|---|---|---|---|---|
| AUC | ACC | AUC | ACC | AUC | ACC |
| $99.7 \pm 0.1$ | $94.0 \pm 0.5$ | $90.8 \pm 0.5$ | $87.7 \pm 1$ | $73.2 \pm 0.2$ | $57.5 \pm 0.3$ |

| Adrenal | | Vessel | | Synapse | |
|---|---|---|---|---|---|
| AUC | ACC | AUC | ACC | AUC | ACC |
| $88.6 \pm 0.1$ | $85.2 \pm 0.4$ | $97.6 \pm 0.7$ | $96.3 \pm 0.3$ | $83.4 \pm 0.7$ | $80.9 \pm 0.2$ |

Table 11: Robustness of HOT performance on timeseries forecasting. The results are obtained from five random seeds.

| Dataset | ECL | | | Weather | | |
|---|---|---|---|---|---|---|
| Horizon | MSE | MAE | SMAPE | MSE | MAE | SMAPE |
| 96 | $0.142 \pm 0.001$ | $0.242 \pm 0.001$ | $0.453 \pm 0.001$ | $0.169 \pm 0.003$ | $0.214 \pm 0.002$ | $0.554 \pm 0.002$ |
| 192 | $0.158 \pm 0.002$ | $0.257 \pm 0.002$ | $0.502 \pm 0.002$ | $0.214 \pm 0.002$ | $0.251 \pm 0.001$ | $0.618 \pm 0.002$ |
| 336 | $0.176 \pm 0.002$ | $0.273 \pm 0.003$ | $0.522 \pm 0.003$ | $0.268 \pm 0.001$ | $0.292 \pm 0.000$ | $0.665 \pm 0.001$ |
| 720 | $0.212 \pm 0.001$ | $0.301 \pm 0.003$ | $0.563 \pm 0.003$ | $0.350 \pm 0.001$ | $0.343 \pm 0.000$ | $0.728 \pm 0.001$ |
| Dataset | Solar | | | Exchange | | |
| Horizon | MSE | MAE | SMAPE | MSE | MAE | SMAPE |
| 96 | $0.194 \pm 0.003$ | $0.229 \pm 0.004$ | $0.365 \pm 0.002$ | $0.083 \pm 0.005$ | $0.202 \pm 0.005$ | $0.180 \pm 0.003$ |
| 192 | $0.227 \pm 0.003$ | $0.254 \pm 0.003$ | $0.394 \pm 0.001$ | $0.173 \pm 0.004$ | $0.295 \pm 0.004$ | $0.278 \pm 0.001$ |
| 336 | $0.244 \pm 0.002$ | $0.267 \pm 0.004$ | $0.417 \pm 0.004$ | $0.313 \pm 0.005$ | $0.405 \pm 0.006$ | $0.396 \pm 0.002$ |
| 720 | $0.252 \pm 0.003$ | $0.274 \pm 0.005$ | $0.431 \pm 0.004$ | $0.804 \pm 0.005$ | $0.673 \pm 0.006$ | $0.644 \pm 0.005$ |
| Dataset | Traffic | | | | | |
| Horizon | MSE | MAE | SMAPE | | | |
| 96 | $0.388 \pm 0.003$ | $0.268 \pm 0.000$ | $0.492 \pm 0.001$ | | | |
| 192 | $0.407 \pm 0.002$ | $0.275 \pm 0.001$ | $0.496 \pm 0.002$ | | | |
| 336 | $0.431 \pm 0.002$ | $0.287 \pm 0.001$ | $0.509 \pm 0.003$ | | | |
| 720 | $0.464 \pm 0.002$ | $0.300 \pm 0.002$ | $0.538 \pm 0.003$ | | | |

Table 12: Full performance comparison between HOT and other baselines on timeseries forecasting with best in **Bold** and second-best in underline. HOT achieves the best performance over the majority of the datasets and all horizons.

| Baselines | | ECL | | Weather | | Traffic | | Solar | | Exchange | |
|---|---|---|---|---|---|---|---|---|---|---|---|
| | | MSE | MAE | MSE | MAE | MSE | MAE | MSE | MAE | MSE | MAE |
| AutoFormer (2021) | 96 | 0.201 | 0.317 | 0.266 | 0.336 | 0.613 | 0.388 | 0.884 | 0.711 | 0.197 | 0.323 |
| | 192 | 0.222 | 0.334 | 0.307 | 0.367 | 0.616 | 0.382 | 0.834 | 0.692 | 0.300 | 0.369 |
| | 336 | 0.231 | 0.338 | 0.359 | 0.395 | 0.622 | 0.337 | 0.941 | 0.723 | 0.509 | 0.524 |
| | 720 | 0.254 | 0.361 | 0.419 | 0.428 | 0.660 | 0.408 | 0.882 | 0.717 | 1.447 | 0.941 |
| SCINet (2022) | 96 | 0.247 | 0.345 | 0.221 | 0.306 | 0.788 | 0.499 | 0.237 | 0.344 | 0.267 | 0.396 |
| | 192 | 0.257 | 0.35 | 0.261 | 0.340 | 0.789 | 0.505 | 0.280 | 0.380 | 0.351 | 0.459 |
| | 336 | 0.269 | 0.369 | 0.309 | 0.378 | 0.797 | 0.508 | 0.304 | 0.389 | 1.324 | 0.853 |
| | 720 | 0.299 | 0.390 | 0.377 | 0.427 | 0.841 | 0.523 | 0.308 | 0.38 | 1.058 | 0.797 |
| FedFormer (2022) | 96 | 0.193 | 0.308 | 0.217 | 0.296 | 0.587 | 0.366 | 0.242 | 0.342 | 0.148 | 0.278 |
| | 192 | 0.201 | 0.315 | 0.276 | 0.336 | 0.604 | 0.373 | 0.285 | 0.380 | 0.271 | 0.315 |
| | 336 | 0.214 | 0.329 | 0.339 | 0.380 | 0.621 | 0.383 | 0.282 | 0.376 | 0.460 | 0.427 |
| | 720 | 0.246 | 0.355 | 0.403 | 0.428 | 0.626 | 0.382 | 0.357 | 0.427 | 1.195 | 0.695 |
| Stationary (2022) | 96 | 0.169 | 0.273 | 0.173 | 0.223 | 0.612 | 0.338 | 0.215 | 0.249 | 0.111 | 0.237 |
| | 192 | 0.182 | 0.286 | 0.245 | 0.285 | 0.613 | 0.340 | 0.254 | 0.272 | 0.219 | 0.335 |
| | 336 | 0.200 | 0.304 | 0.321 | 0.338 | 0.618 | 0.328 | 0.290 | 0.296 | 0.421 | 0.476 |
| | 720 | 0.222 | 0.321 | 0.414 | 0.410 | 0.653 | 0.355 | 0.285 | 0.295 | 1.092 | 0.769 |
| TiDE (2023) | 96 | 0.237 | 0.329 | 0.202 | 0.261 | 0.805 | 0.493 | 0.312 | 0.399 | 0.094 | 0.218 |
| | 192 | 0.236 | 0.330 | 0.242 | 0.298 | 0.756 | 0.474 | 0.339 | 0.416 | 0.184 | 0.307 |
| | 336 | 0.249 | 0.344 | 0.287 | 0.335 | 0.762 | 0.477 | 0.368 | 0.430 | 0.349 | 0.431 |
| | 720 | 0.284 | 0.373 | 0.351 | 0.386 | 0.719 | 0.449 | 0.370 | 0.425 | 0.852 | 0.698 |
| Crossformer (2023) | 96 | 0.219 | 0.314 | **0.158** | 0.230 | 0.522 | 0.290 | 0.310 | 0.331 | 0.256 | 0.367 |
| | 192 | 0.231 | 0.322 | 0.206 | 0.277 | 0.530 | 0.293 | 0.734 | 0.725 | 0.470 | 0.509 |
| | 336 | 0.246 | 0.337 | 0.272 | 0.335 | 0.558 | 0.305 | 0.750 | 0.735 | 1.268 | 0.883 |
| | 720 | 0.280 | 0.363 | 0.398 | 0.418 | 0.589 | 0.328 | 0.769 | 0.765 | 1.767 | 1.068 |
| RLinear (2023) | 96 | 0.201 | 0.281 | 0.192 | 0.232 | 0.649 | 0.389 | 0.322 | 0.339 | 0.093 | 0.217 |
| | 192 | 0.201 | 0.28 | 0.240 | 0.271 | 0.601 | 0.366 | 0.359 | 0.356 | 0.184 | 0.307 |
| | 336 | 0.215 | 0.298 | 0.292 | 0.307 | 0.609 | 0.369 | 0.397 | 0.369 | 0.351 | 0.432 |
| | 720 | 0.257 | 0.331 | 0.364 | 0.353 | 0.647 | 0.387 | 0.397 | 0.356 | 0.886 | 0.714 |
| DLinear (2023) | 96 | 0.197 | 0.282 | 0.196 | 0.255 | 0.650 | 0.396 | 0.290 | 0.378 | 0.088 | 0.218 |
| | 192 | 0.196 | 0.285 | 0.237 | 0.296 | 0.598 | 0.370 | 0.320 | 0.398 | 0.176 | 0.315 |
| | 336 | 0.209 | 0.301 | 0.283 | 0.335 | 0.605 | 0.373 | 0.353 | 0.415 | 0.313 | 0.427 |
| | 720 | 0.245 | 0.333 | **0.345** | 0.381 | 0.645 | 0.394 | 0.356 | 0.413 | 0.839 | 0.695 |
| TimesNet (2023) | 96 | 0.168 | 0.272 | 0.172 | 0.220 | 0.593 | 0.321 | 0.250 | 0.292 | 0.107 | 0.234 |
| | 192 | 0.184 | 0.289 | 0.219 | 0.261 | 0.617 | 0.336 | 0.296 | 0.318 | 0.226 | 0.344 |
| | 336 | 0.198 | 0.300 | 0.280 | 0.306 | 0.629 | 0.336 | 0.319 | 0.330 | 0.367 | 0.448 |
| | 720 | 0.220 | 0.320 | 0.365 | 0.359 | 0.640 | 0.350 | 0.338 | 0.337 | 0.964 | 0.746 |
| PatchTST (2023) | 96 | 0.181 | 0.270 | 0.177 | 0.218 | 0.462 | 0.295 | 0.234 | 0.286 | 0.088 | 0.205 |
| | 192 | 0.188 | 0.274 | 0.225 | 0.259 | 0.466 | 0.296 | 0.267 | 0.310 | 0.176 | 0.299 |
| | 336 | 0.204 | 0.293 | 0.278 | 0.297 | 0.482 | 0.304 | 0.290 | 0.315 | **0.301** | **0.397** |
| | 720 | 0.246 | 0.324 | 0.354 | 0.348 | 0.514 | 0.322 | 0.289 | 0.317 | 0.901 | 0.714 |
| iTransformer (2024) | 96 | 0.148 | **0.240** | 0.174 | 0.221 | 0.395 | **0.268** | 0.203 | 0.237 | 0.086 | 0.206 |
| | 192 | 0.162 | **0.253** | 0.221 | 0.254 | 0.417 | 0.276 | 0.233 | 0.261 | 0.177 | 0.299 |
| | 336 | 0.178 | **0.269** | 0.278 | 0.296 | 0.433 | 0.283 | 0.248 | 0.273 | 0.331 | 0.417 |
| | 720 | 0.225 | 0.317 | 0.358 | 0.347 | 0.467 | 0.302 | **0.249** | 0.275 | 0.847 | 0.691 |
| HOT (Ours) | 96 | **0.142** | 0.242 | 0.169 | **0.214** | **0.388** | **0.268** | **0.194** | **0.229** | **0.083** | **0.202** |
| | 192 | **0.158** | 0.257 | **0.214** | 0.251 | **0.407** | 0.275 | **0.227** | **0.254** | 0.173 | **0.295** |
| | 336 | **0.176** | 0.273 | 0.268 | **0.292** | **0.431** | 0.287 | **0.244** | **0.267** | 0.313 | 0.405 |
| | 720 | **0.212** | **0.301** | 0.350 | **0.343** | **0.464** | **0.300** | 0.252 | **0.274** | **0.804** | **0.673** |

