# OpenReview forum: "Higher Order Transformers: Efficient Attention Mechanism for Tensor Structured Data"
_ICLR.cc/2025/Conference — Submitted to ICLR 2025_

### Official Review · Reviewer_ccUZ · 2024-10-18

**Soundness:** 2
**Presentation:** 2
**Contribution:** 1
**Rating:** 3
**Confidence:** 4

**Summary:**

The authors propose a high-order Transformer (HOT) for high-order tensor input. The authors apply Kronecker tensor decomposition and kernel attention in Performer to propose a novel Kronecker factorized attention. The experimental results in time-series forecasting and 3D medical image classification illustrate the efficient of HOT.

**Strengths:**

1. The authors skillfully perform tensor operations.

**Weaknesses:**

1. This paper demonstrates a limited degree of innovation. The proposed methodology in this paper is merely based on Kronecker tensor decomposition [1] and kernel attention [2].

[1] Phan, K. (2012). On Revealing Replicating Structures in Multiway Data: A Novel Tensor Decomposition Approach. In Latent Variable Analysis and Signal Separation (pp. 297–305). Springer Berlin Heidelberg.

[2] Krzysztof Marcin Choromanski, Valerii Likhosherstov, David Dohan, Xingyou Song, Andreea Gane, Tamas Sarlos, Peter Hawkins, Jared Quincy Davis, Afroz Mohiuddin, Lukasz Kaiser, David Benjamin Belanger, Lucy J Colwell, \& Adrian Weller (2021). Rethinking Attention with Performers. In International Conference on Learning Representations.

**Questions:**

1. How does the use of low-rank approximation via Kronecker decomposition in Section 4.2 impact the trade-off between computational complexity and precision? The authors need to provide more analysis on this and offer guidelines for choosing the appropriate rank R or kernels to help practitioners adapt the method to different datasets and needs.

2. The authors need to provide more insights or proof on how HOT maintains low complexity when handling datasets with more than 3 dimensions. It would be helpful to understand how the model scales and whether the current complexity reduction techniques, like Kronecker decomposition remain effective as the dimensionality increases.

---

> ### Author Response · Authors · 2024-11-20
> **Response to Reviewer ccUZ**
>
> We thank the reviewer for the valuable feedback and acknowledging our contributions and the effectiveness of the proposed method. We especially thank the reviewer for pointing out the unclarity in the writing about the rank analysis and the choice of attention kernel. First we address the concern about the novelty of our work:
>  - Regarding the argument about the novelty and innovation in our work, although both techniques used in our method, Kronecker decomposition and kernel attention, have already been introduced earlier and widely used, they have never been studied in the context of efficient yet expressive high order attention. Moreover, we provided the general formulation, mathematical formalism, and theoretical guarantee of high order attention any high-order input data with no specific assumptions about the size and structure of the data. Furthermore, we provided comprehensive comparisons with baselines, ablation studies of various aspects of the proposed method, and efficiency analysis on two higher-order tasks, namely multivariate timeseries forecasting and 3D medical image classification.
>
>
> Now, we answer the questions asked by the reviewer:
>
>
>  - Question 1: As shown in Section 5.3.1, we studied the effect of different numbers of attention heads (a.k.a. attention rank) for both tasks. Although the effective rank for the Kronecker decomposition varies for different tasks and datasets and is highly dependent on the data size and its characteristics, in practice, we see that low-rank approximation (where the rank or the number of heads is not a large number) is expressive enough to yield good results. To that end, we treat the rank as a hyper-parameter of the model and choose it by hyper-parameter search based on validation metrics for each dataset. This approach makes it very easy to achieve a proper performance, without any need for exhaustive analysis on the data. For the choice of attention kernel, we fixed the feature map function to SMReg from the Performer paper for all our models, as it was shown to be the most stable, fast converging and expressive by the authors. It is worth mentioning that we have added Section 5.3.3 in the revised paper to study the effect of the proposed factorization and using the kernel trick on the performance and efficiency. As shown in Tables 4 and 5, with proper hyper-parameter tuning (including on rank), HOT can yield as good results as the full attention with much less memory footprint and training time.
>
>
>  - Question 2: As shown in Section 4.3 the computational complexity of HOT is $\mathcal{O}(D^2\Pi N_i)$, where D is the attention dimension and $N_i$s are the input tensor dimensions, scaling linearly with respect to the input dimensions and quadratic with respect to the model’s hidden dimension.
>
>
> We have updated Section 5.3.3 and Implementation Details in the appendix to address the ambiguity pointed out in Q1. We hope we have addressed all your concerns, in which case we kindly ask you to consider increasing your score. We remain of course available to answer any further questions.

---

### Official Review · Reviewer_B21D · 2024-10-29

**Soundness:** 3
**Presentation:** 3
**Contribution:** 2
**Rating:** 3
**Confidence:** 5

**Summary:**

This work proposed a transformer architecture to efficiently deal with higher-order structure data.

**Strengths:**

1. The research question is important since the attention operator is very computationally expensive on higher-order data.
2. The usage of Kronecker to factorize attention is sound.
3. Empirical studies demonstrate the effectiveness of proposed work on higher-order data.

**Weaknesses:**

1. The major problem is the novelty. This work is essentially about attention instead of transformer. The Kronecker factorization is applied to attention, which is part of the transformer architecture. Actually, applying Kronecker factorization to attention on higher-order data has been explored in [1]. Currently, the novelty is very marginal from my current understanding. It is important for authors to demonstrate the distinct contributions from [1].

2. Since the proposed attention is not necessarily integrated with the transformer architecture, it would be better to show its effectiveness in other architectures, such as Attention U-Net [2].

3. [Added after discussion with authors] The evaluation is actually conducted on 1D data. The time series data is reshaped into 2D to mimic higher-order data. However, this does not make sense since there is no locality in the feature dimension of 1D data. It does not make sense to apply this on 1D data although it is mathematically correct. To improve, the evaluation on real 2D and 3D data is preferred instead of using 1D data.

[1] Gao, Hongyang, Zhengyang Wang, and Shuiwang Ji. "Kronecker attention networks." In Proceedings of the 26th ACM SIGKDD International Conference on Knowledge Discovery & Data Mining, pp. 229-237. 2020.
[2] Oktay, Ozan, Jo Schlemper, Loic Le Folgoc, Matthew Lee, Mattias Heinrich, Kazunari Misawa, Kensaku Mori et al. "Attention u-net: Learning where to look for the pancreas." arXiv preprint arXiv:1804.03999 (2018).

**Questions:**

Please see weakness.

---

> ### Author Response · Authors · 2024-11-20
> **Response to Reviewer B21D**
>
> We thank the reviewer for the valuable feedback and specially appreciate the reviewer for acknowledging the importance of the research question. First we address the concern about the novelty of our work:
>
>  - Regarding the argument about the novelty and innovation in our work, although both techniques used in our method, Kronecker decomposition and kernel attention, have already been introduced earlier and widely used, they have never been studied in the context of efficient yet expressive high order attention. Moreover, we provided the general formulation, mathematical formalism, and theoretical guarantee of high order attention any high-order input data with no specific assumptions about the size and structure of the data. Furthermore, we provided comprehensive comparisons with baselines, ablation studies of various aspects of the proposed method, and efficiency analysis on two higher-order tasks, namely multivariate timeseries forecasting and 3D medical image classification.
>
>
> Now, we address the weaknesses as follows:
>
> - Both of the papers propose an attention mechanism to reduce
> the memory and computation complexity of attention networks applied to high order data by using some notion of tensor operation in their formulation. However, there are several key differences:
>     - KAN [1] assumes a 2D structure and a multivariate normal distribution for the input tensor and models the covariance using a kronecker sum of covariance matrices for rows and columns. HOT, on the other hand, does not make any prior assumptions on the distribution or the tensor structure of the input data and it can be applied to any-order multi-modal tensors.
>    - KAN does not actually perform any Kronecker Decompositions or Kronecker Products in practice. Instead, it proposes to take the mean over the rows and columns of query/key/value matrices instead of their vectorized version which is at the end equivalent to the kronecker sum. On the other hand, HOT utilises a rank R Kronecker Decomposition of the high order full attention, making it easy to control the complexity-expressivity tradeoff and interpret first-order attention matrices.
>
>     In conclusion, even though their names are very similar the two methods are drastically different, hence the novelty of our contribution is not diminished by this previous work. We thank the reviewer for bringing this reference to our attention, we added a citation and discussion of this work in the Related Work section.
>
>  - Although the use of the proposed factorized linear attention was only studied in the context of the Transformers architecture, its capabilities in other architectures such as Attention U-net [2] is yet to be explored. We have added this as one of the possible future directions of our work. As mentioned in the conlusion, we believe that high-order attention can be a powerful alternative to the current attention modules used in image and video diffusion models (specifically within the U-Net module) to enhance the spatial and temporal coherence of the outputs.
>
> We hope we have addressed all your concerns, in which case we kindly ask you to consider increasing your score. We remain of course available to answer any further questions.
>
> [1] Gao, Hongyang, Zhengyang Wang, and Shuiwang Ji. "Kronecker attention networks." In Proceedings of the 26th ACM SIGKDD International Conference on Knowledge Discovery & Data Mining, pp. 229-237. 2020.
>
>
> [2] Oktay, Ozan, Jo Schlemper, Loic Le Folgoc, Matthew Lee, Mattias Heinrich, Kazunari Misawa, Kensaku Mori et al. "Attention u-net: Learning where to look for the pancreas." arXiv preprint arXiv:1804.03999 (2018).

---

> > ### Comment · Reviewer_B21D · 2024-11-22
> >
> > Thank you for the detailed rebuttal.
> >
> > Regarding Weakness 1, I agree that MAN is primarily elaborated on 2D data, though it has the potential to extend to higher-order data. However, it is essential to compare with MAN using 2D data to verify whether the assumptions made in MAN hold in practice and assess their impact on model performance.
> >
> > For Weakness 2, the lack of testing on other model architectures limits the evaluation of the proposed approach's soundness across broader application scenarios.
> >
> > Based on the above, I would like to keep my score.

---

> ### Author Response · Authors · 2024-11-26
> **Response to Reviewer B21D**
>
> Thank you for your follow-up comment.
>
> Regarding weakness 1, first, we would like to provide a deeper comparison between KAN and HOT on the theoretical level:
>
>  - KAN models the covariance matrix of the input data using a Kronecker sum operation, assuming that the rows and columns of the input data follow matrix-variate normal distribution which may not be necessarily accurate in most cases and also imposes independence between rows and columns. In practice, KAN reduces the full attention matrix $S \in \mathbb{R}^{N_1 N_2 … N_k \times N_1 N_2 … N_k}$ to an attention matrix $\tilde{S} \in \mathbb{R}^{(N_1+N_2+…+N_k) \times (N_1+N_2+…+N_k)}$ if extended to higher orders. This dramatic reduction, while makes KAN very efficient, loses a lot of information and prevents the attention from capturing any high-order interactions between axes more than first and second order even when applied to data with higher orders. Thus, KAN does not naturally extend due to its reliance on 2D assumptions. On the other hand, HOT relies on theoretical guarantee of rank R Kronecker decomposition while also incorporating RoPE on every axis to preserve structural and positional information. It can also balance expressivity and scalability by allowing flexible trade-offs through the rank.
>
>  - KAN and HOT can be implemented with the same number of parameters and yield the same computational complexities.
>
> On the experimental level, while providing performance comparison with KAN on the Imagenet-1k dataset is not feasible for us within the short time window of the discussion period, we will report HOT performance on this benchmark in the final revision. However, we have provided partial comparison of HOT and KAN on three timeseries datasets **(as 2D datasets)** in the following. These results are obtained using [this](https://github.com/lucidrains/kronecker-attention-pytorch) implementation and integrating it into the same model configuration used in all our experiments:
>
> |  | Weather | | | Exchange | | | Solar | |
> | -------- | ------- | --------| - | --------|-------- | - | ------- | -------- |
> | | MSE | MAE |.  | MSE | MAE |  | MSE | MAE|
> |  **KAN**  |   0.262  | 0.291 |    | 0.430 | 0.450 |     | 0.273 | 0.300 |
> |  **HOT** |  **0.250** | **0.275**|   | **0.343** | **0.394**|    | **0.229** | **0.256**   |
>
>
> Regarding Weakness 2, we would like to emphasize that the scope of this work was intentionally constrained to the Transformer architecture for the following reasons:
>  - Transformers offer a reliable and modular baseline for systematically testing new attention mechanisms. Using Transformers ensures consistent experimental conditions, enabling a clear assessment of the proposed Kronecker factorized attention's performance, scalability, and efficiency.
>  - Transformers are highly versatile and have been successfully applied across multiple domains, including NLP, computer vision, and time-series analysis. This versatility makes them an ideal choice to showcase the utility of our proposed method on tasks such as multivariate time series forecasting and 3D medical image classification where best performing methods also follow Transformers architecture.
> - Validating Kronecker factorized attention within the Transformer architecture establishes a groundwork for extending this mechanism to other models, such as Attention U-Net, as suggested, or further specialized architectures in areas like generative modeling or biomedical applications.
>
>
> We hope we have addressed all your concerns, in which case we kindly ask you to consider increasing your score. We remain of course available to answer any further questions.

---

> > ### Comment · Reviewer_B21D · 2024-12-02
> >
> > Thanks for the effort of providing comparison results with KAN. Can you explain more settings about the experiment, especially how to use three timeseries datasets as 2D datasets?

---

> > > ### Author Response · Authors · 2024-12-02
> > > **Response to Reviewer B21D**
> > >
> > > Thank you for your follow-up comment.
> > >
> > > We implemented a transformer model with the same architecture as ours, except using KAN attention instead of our proposed factorized attention. We performed the same hyperparameter-tuning strategy as ours to keep the comparison fair while the only changing factor is the attention module.
> > >
> > > Following the explanation in the original paper and the aforementioned Github repository, KAN takes the average over each axis and applies scaled dot-product attention on the concatenates of the two. Resulting in a query, key and value matrices of size (batch, N + T, d) where N and T are the number of variates and timesteps respectively. Lastly, it reshapes the output back to the original dimensions using outer sum operation.
> > >
> > > We hypothesize that the inferior performance of KAN compared to HOT on the Weather, Exchange, and Solar datasets is due to two main factors:
> > >  - The averaging done on each axis is a suboptimal way of dimension reduction, causing information loss in the attention module. On the other hand, HOT controls the efficiency/expressivity trade-off with a rank R decomposition.
> > >  - KAN, assumes a normal distribution on the input matrix, while it may not be the case for most datasets. In contrast, HOT does not make any specific assumptions on the number of dimensions, or data distribution, carrying much less inductive bias.

---

> > > > ### Comment · Reviewer_B21D · 2024-12-03
> > > >
> > > > My question is essentially how to apply HOT and KAN on time series data since they are 1-D data. Maybe I miss some details.

---

> > > > > ### Author Response · Authors · 2024-12-04
> > > > > **Response to Reviewer B21D**
> > > > >
> > > > > As highlighted in the task description in Section 5.1, the input to the model would be the historical observations over all variables within a time window resulting in a 2D matrix $X \in\mathbb{R}^{N\times T}$ with $T$ time steps and $N$ variates. We assume the number of features (a.k.a. channels) is 1.

---

### Official Review · Reviewer_Gjkf · 2024-11-06

**Soundness:** 2
**Presentation:** 3
**Contribution:** 3
**Rating:** 6
**Confidence:** 2

**Summary:**

This paper proposes a novel Transformer architecture to extend traditional Transformers for processing higher-order data, while reducing the attention cost to quadratic in each axis' dimension and the complexity to linear.

**Strengths:**

* Considering the importance of higher-order data, designing specialized Transformer architectures for higher-order data is meaningful.

* The proposed Higher-Order Transformers (HOT) can efficiently capture essential cross-dimensional dependencies with lower complexity, thus overcoming the limitations of previous methods.

* The experimental results also support the superiority of the proposed HOT, both in effectiveness and efficiency.

**Weaknesses:**

* Different form previous works which directly flatten the input tensor into a sequence, it's impossible for HOT to use the same architecture process data with different orders, thus limiting its scalability.

* Consideing the high-order feature of HOT, it's necessary to compare the model parameters with its baselines.

* Besides, it's also necessary to conduct more experiments to demonstrate the generality of the proposed method on various kinds of higher-order data, such as typical 2D image datasets and video dataset.

**Questions:**

Please see the **weaknesses**.

---

> ### Author Response · Authors · 2024-11-20
> **Response to Reviewer Gjkf**
>
> We thank the reviewer for the valuable comments on our paper and appreciating the contribution and effectiveness of our proposed method. Below we answer the questions raised by the reviewer:
>
> - Questions 1, and 2: The architecture and the number of model parameters stay the same for each dataset regardless of the attention order and other configurations as shown in ablation studies in Section 5.3.2. Additionally, we have added Section 5.3.3 in the paper to compare the factorized attention with the naive full attention (aka. flattened attention). This study shows that HOT is as easy to use and scalable as the flattening approach while being much more efficient and less memory intensive while using the same model weights. We have also added a new column to the Tables 1 and 2 to compare the number of parameters with the baselines for both tasks and HOT is always among the models with the fewest parameters.
>
>  - Question 3: As already mentioned in the conclusion and future direction, we acknowledge that conducting more experiments on other high-order datasets including image and video could showcase even more the effectiveness of HOT. While it is not feasible for us to provide results on such datasets within the short time window of the discussion period, we will, however, explore HOT performance on ImageNet-1K dataset compared to popular baselines such as ViT in the final revision.
>
>
> We hope we have addressed all your concerns, in which case we kindly ask you to consider increasing your score. We remain of course available to answer any further questions.

---

### Official Review · Reviewer_2H8G · 2024-11-07

**Soundness:** 3
**Presentation:** 3
**Contribution:** 2
**Rating:** 3
**Confidence:** 2

**Summary:**

This paper proposes a new transformer-based model, i.e., Higher-Order Transformers (HOT) which can efficiently process data with more than two axes (i.e., higher order tensors). In addition, the authors propose a novel Kronecker factorized attention mechanism that reduces the attention cost and provide theoretical proof of Kronecker decomposition. The experiments over multivariate time series data and 3D medical images are straightforward and clear.

**Strengths:**

+ The paper is clear and well written. It is easy to follow and presents a concise idea of Kronecker factorized attention mechanism based on Kronecker decomposition.
+ The paper also do a good job in describing existing research/roadmap in transformer-based models.
+ For multivariate forecasting tasks, the proposed HOT model always outperforms baselines.

**Weaknesses:**

- For multivariate forecasting tasks, the model's effectiveness is assessed only through MAE and MSE. Can the authors also report other metrics, e.g., MAPE? In addition, I am curious about the confidence intervals for prediction of the HOT model compared to other baselines.
- Why Kronecker decomposition: there are also other higher-order structure decomposition approach, e.g., CP decomposition, Tucker decomposition, etc. I wonder if the authors have compared Kronecker decomposition with other tensor decomposition methods?
- Improvements are minor: although the HOT model achieves the best prediction accuracy across all datasets for multivariate forecasting tasks, improvements are not significant. Also, for 3D medical images, the HOT model's performances are less competitive.
- It would be helpful if the authors could conduct robust experimental comparisons, i.e., if the proposed Kronecker factorized attention mechanism can help improve model's robustness.

**Questions:**

See my questions from the Weaknesses.

---

> ### Author Response · Authors · 2024-11-20
> **Response to Reviewer 2H8G**
>
> We thank the reviewer for the valuable feedback and acknowledge our technical contributions and the effectiveness of the proposed method. We address the concerns raised by the reviewer as follows:
>
>  - Question 1:  As an addition to MAE and MSE we have added the reported values for SMAPE or symmetric MAPE in Table 11 in the appendix as an alternative to MAPE to make the results more interpretable. The are two main reasons for this choice:
>     - MAPE treats positive and negative errors asymmetrically, potentially leading to biased interpretations, especially in long-range forecasting where both over- and under-predictions are important.
>     - When actual values are zero or close to zero, MAPE can produce unstable or out-of-bound values..
>
> We are currently working on adding confidence intervals visualisations and we will post a comment with updated results as soon as they are available.
>
>  - Question 2: The Kronecker decomposition we use for HOT is actually equivalent to a CP decomposition, this is the key element of the proof of our theorem. More precisely, the CP decomposition of the attention matrix reshaped into a tensor of size $N_1^2\times N_2^2 \times \cdots \times N_k^2$ is **exactly equivalent** to the Kronecker factorization of the attention matrix we use for HOT. We added a discussion to clarify this point in section B in the appendix after the proof.
> Other decomposition approaches such as Tucker, Tensor-train, etc. yield different structures that could be investigated as future work but are out of scope of the present study.
> The choice of Kronecker decomposition was initially motivated by three main reasons:
>     - Simplicity and nice properties that enable us to link it easily to the first-order attention matrices that are already ubiquitous and easy to interpret.
>     - Approximation guarantee when the rank increases.
>     - Integrates well with the kernel trick to make attention linear in complexity over all axes while maintaining flexiblity.
>
>
> - Question 3: We adopted the metric computation function for 3D medical image classification task from the MedMNIST python package to be consistent with the rest of the baselines and updated the results accordingly in Tables 2 and 4. This change slightly improved AUC over multiple datasets and made HOT, the best performing model on Organ, Fracture, Adrenal, and Vessel datasets and the second best on the Synapse dataset. Additionally, it is worth mentioning that the main promise of HOT is to give acceptable performance while being flexible, scalable, and highly efficient compared to the rest of the baselines.
>
> - Question 4: We have added the robustness analysis of HOT over both tasks in the appendix in Tables 10 and 11. The results were obtained from five different random seeds.
>
>
> We hope we have addressed all your concerns, in which case we kindly ask you to consider increasing your score. We remain of course available to answer any further questions.

---

> > ### Comment · Reviewer_2H8G · 2024-11-24
> > **Reply to authors feedback**
> >
> > Thank you the authors for addressing my points here - I think the responses have helped the clarity of the paper. I understand the motivation of the Kronecker decomposition is clear, however, it is unclear the difference (e.g., numerical performance difference) among different decomposition methods. I will maintain my score.

---

> ### Author Response · Authors · 2024-11-26
> **Response to Reviewer 2H8G**
>
> We appreciate your acknowledgment of the improved clarity and theoretical foundation of our approach. While we recognize the importance of exploring various tensor decomposition methods, the choice of Kronecker decomposition in this work was motivated by its simplicity, scalability, and expressivity. A comprehensive numerical comparison would require additional experimentation beyond the scope of this study. However, we plan to investigate these methods in future work to explore their potential impact on performance and efficiency. For now, to further justify our choice of Kronecker decomposition, we provide a detailed analysis of how other well-known tensor factorization methods could be integrated into higher-order attention and their trade-offs compared to our proposed factorization.
>
> ### CP Decomposition
>
> As highlighted in Section B of the appendix, CP decomposition is mathematically equivalent to the proposed Kronecker decomposition. Specifically:
> $S_{\text{attention}} = \sum_{r}^R S_r^{(1)} \otimes S_r^{(2)} \otimes \cdots \otimes S_r^{(k)}$
> where $S_r^{(i)} \in \mathbb{R}^{N_i \times N_i} $ represents the first-order attention matrix along axis $ i $.
>
> - **Number of parameters**: $ \mathcal{O}(D^2) $
> - **Computational complexity**: $ \mathcal{O}(D (\sum N_i) \prod N_i) $
>
> Since the parameterization and complexity of CP and Kronecker decompositions are identical in our setting, CP decomposition does not provide additional benefits in terms of expressivity or efficiency.
>
>
> ### Tucker Decomposition
> For Tucker decomposition with a fixed rank $R $ for all axes, the attention matrix is expressed as:
>
> $S_{\text{attention}} = \sum_{i_1, i_2, ..., i_k}^R \mathcal{G_{i_1, i_2, ..., i_k}}  S_{i_1}^{(1)} \otimes S_{i_2}^{(2)} \otimes \cdots \otimes S_{i_k}^{(k)}$
>
> where $S_{i_j}^{(j)} \in \mathbb{R}^{N_j \times N_j}$ is the first-order attention matrix for axis $i$, and $\mathcal{G} \in \mathbb{R}^{R^k} $ is a core tensor that enhances expressivity by introducing additional parameters.
>
> While the addition of the core tensor $ \mathcal{G} $ increases expressivity, it significantly impacts scalability and efficiency:
> - **Number of parameters**: $ \mathcal{O}(D^2 + R^k) $
> - **Computational complexity**: $ \mathcal{O}(D (R^k + \sum N_i) \prod N_i) $
>
> The exponential growth in parameters and complexity with $R^k $ makes Tucker decomposition less suitable for highly scalable and efficient models, especially as the order $k $ of the tensor increases.
>
> ### Tensor Train Decomposition
> For tensor train decomposition with a fixed rank \( R \) for all factor matrices, the attention matrix is expressed as:
> $
> S_{\text{attention}} = \sum_{i_1, i_2, \ldots, i_{k-1}}^R S_{i_1}^{(1)} \otimes S_{i_1, i_2}^{(2)} \otimes S_{i_2, i_3}^{(3)} \otimes \cdots \otimes S_{i_{k-1}}^{(k)},
> $
> where $ S^{(i)} \in \mathbb{R}^{R \times N_i \times N_i} $ for $ i = 1, k $, and $S^{(i)} \in \mathbb{R}^{R \times R \times N_i \times N_i} $ for $ i \in [2, k-1] $.
>
> While tensor train decomposition adds only a small overhead in computational complexity:
> - **Number of parameters**: $ \mathcal{O}(D^2) $ or potentially higher due to factor tensors.
> - **Computational complexity**: $ \mathcal{O}(D (R^2 + \sum N_i) \prod N_i) $.
>
> However, constructing fourth-order factor tensors $S^{(i)} \in \mathbb{R}^{R \times R \times N_i \times N_i} $ is non-trivial and may require additional parameters or modifications to align with our use case. This lack of simplicity makes tensor train decomposition less appealing compared to Kronecker decomposition.
>
>
>
>
> To conclude, the choice of Kronecker decomposition in our work is motivated by its:
> 1. **Simplicity**: It avoids additional overhead, such as core tensors or complex factor tensors, making it easy to implement and interpret.
> 2. **Efficiency**: Kronecker decomposition scales efficiently with the tensor order $ k $, avoiding exponential growth in parameters or computations.
> 3. **Expressivity**: It balances expressivity and scalability by allowing flexible trade-offs through the rank $ R $.
>
> Moreover, our empirical evaluations confirm the effectiveness of this approach, achieving strong performance across tasks while maintaining computational efficiency. We hope this analysis provides further clarity and justifies our methodological choice, in which case we kindly ask you to consider increasing your score. We remain of course available to answer any further questions.

---

### Author Response · Authors · 2024-11-20
**General Response to All Reviewers**

We sincerely thank all the reviewers for their time and insightful comments. We are excited that all of the reviewers acknowledged the effectiveness and simplicity of the proposed method, and we appreciate that they found our paper is well motivated (Reviewers **Gjkf, B21D**), well presented (Reviewers **B21D, Gjkf, 2H8G**), and mathematically sound (Reviewer **B21D, ccUZ**).


To the best of our efforts, we have provided detailed comments to address all concerns raised by each reviewer. Meanwhile, we have carefully revised the paper and appendices where the main modifications are highlighted in blue. Specifically, the main revisions we made are as follows:

 - Updated Section 5.3.1 by adding explanation about the choice of rank and the effectiveness of the low-rank approximation.
 - Updated the AUC scores for 3D medical image classification results in Tables 2 and 4 using the same function as the original baselines to make the results more consistent for fair comparison.
 - Added Section 5.3.3 to study the effect of using the proposed Kronecker factorization and the attention kernel trick on the performance and efficiency.
 - Updated the efficiency analysis in Figure 4 based on the experiments ran on an Nvidia A100 GPU.
 - Added Robustness Analysis of the HOT performance on both tasks in the Tables 10 and 11 in the appendix.
 - Reported SMAPE as additional metric on timeseries forecasting in Table 11.
 - Added a comparison on the number of parameters with the baselines in Tables 1 and 2.

---

### Meta-Review · Area_Chair_yD35 · 2024-12-18

**Metareview:**

In this submission, the authors introduce a high-order Transformer associated with a factorization-based acceleration strategy. However, the reviewers have concerns about the experimental settings and the performance of the proposed model. In particular, introducing high-order decomposition may increase runtime and memory costs significantly, but the authors did not provide comparisons for various methods on their runtime and memory costs. Compared with the increase in computations, whether the improvements are significant is questionable. Although the authors made efforts to solve the concerns of the reviewers, the reviewers still think the authors need to enhance the experimental part, and accordingly, the submission requires a next-round review.

**Additional Comments On Reviewer Discussion:**

In the rebuttal-discussion phase, the reviewers interacted with the authors. The authors' rebuttals did not fully resolve the reviewers' concerns. After discussing with the reviewers, I decided to reject this submission.

---

### Decision · Program_Chairs · 2025-01-22

Reject